# Hallo3D: Multi-Modal Hallucination Detection and Mitigation for Consistent 3D Content Generation

**Hongbo Wang**[1,2]   **Jie Cao**[1,2]   **Jin Liu**[1,3]   **Xiaoqiang Zhou**[1,4]   **Huaibo Huang**[1,2*] **Ran He**[1,2]

[1]MAIS & NLPR, Institute of Automation, Chinese Academy of Sciences, Beijing, China
[2]School of Artificial Intelligence, University of Chinese Academy of Sciences, Beijing, China
[3]School of Information Science and Technology, ShanghaiTech University, Shanghai, China
[4]University of Science and Technology of China, Hefei, China
`wanghongbo2024@ia.ac.cn`, `jie.cao@cripac.ia.ac.cn`, `liujin2@shanghaitech.edu.cn`
`xq525@mail.ustc.edu.cn`, `huaibo.huang@cripac.ia.ac.cn`, `rhe@nlpr.ia.ac.cn`

## Abstract

Recent advances in pretrained 2D diffusion models have significantly improved visual prior guidance for 3D content generation. However, this process often lacks geometric constraints, leading to spatial perception hallucinations and multi-view inconsistencies. To address this, we introduce **Hallo3D**, a tuning-free method for 3D content generation that leverages the geometric perception capabilities of large multi-modal models to detect and mitigate these hallucinations. Our approach follows a generation-detection-correction paradigm, using multi-modal inconsistencies as query information to guide the detection of hallucinations and formulate enhanced negative prompts that ensure consistent renderings. Additionally, we propose a denoising strategy that employs attention mechanisms to maintain consistency in color and texture across multiple views during visual guidance. Our method is data-independent, easily integrates with existing 3D content generation frameworks, and supports both text-driven and image-driven approaches. Extensive experiments demonstrate that our method significantly improves the consistency and quality of generated 3D content, particularly in mitigating hallucinations common with 2D pretrained models.

## 1 Introduction

Recent studies on 3D content generation have made significant progress, emerging as a central research focus in computer vision and computer graphics. The approaches for 3D content generation can be categorized into two primary categories: those based on 2D priors and those based on 3D priors. The strategies utilizing 2D priors typically learn 3D representations by approximating the probability distribution of 2D rendered images relative to a pre-trained diffusion model. This approximation is achieved during the visual guidance phase through a sophisticated optimization technique known as Score Distillation Sampling (SDS) [40].

However, methods based on 2D priors often suffer from overfitting to specific viewpoints of rendered images, resulting in generated 3D content that deviates from the expected distribution [2]. This overfitting leads to spatial perception inaccuracies, such as the Janus problem, where the generated objects display implausible duplications of features like faces or limbs, as depicted in Fig.1. An intuitive would be to learn priors from high-quality 3D data [19, 36, 50, 65, 62, 29, 18]. However, the limited availability and the often sparse supervision of 3D data pose significant challenges to maintaining view consistency and enhancing the generalizability of the generated content [23, 21]. Moreover, due to 3D

---

*Huaibo Huang is the corresponding author.

38th Conference on Neural Information Processing Systems (NeurIPS 2024).

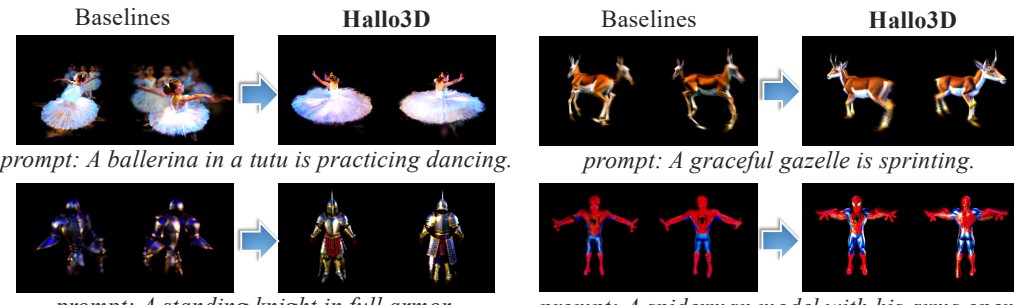

| Baselines | **Hallo3D** | Baselines | **Hallo3D** |

*prompt: A ballerina in a tutu is practicing dancing.*     *prompt: A graceful gazelle is sprinting.*

*prompt: A standing knight in full armor.*     *prompt: A spiderman model with his arms open.*

Figure 1: 3D Content Generation Results between Hallo3D (ours) and Baseline Model. Hallo3D can effectively solve the "Janus" problem and improve the multi-view consistency of the 3D generation.

generation tasks spanning a diverse array of domains, the scalability of data-driven models remains markedly constrained, limiting their applicability across a comprehensive spectrum of potential uses.

To mitigate the hallucination problem and ensure view-consistent generation, we leverage the large multi-modal models to infer and adjust the geometric structures of the generated content. These models can recognize spatial relationships and evaluate the structural consistency of visual contexts by interpreting 3D elements such as lighting and proportion from 2D renderings. Building on this observation, we propose a novel generation-detection-correction paradigm. In this paradigm, we utilize multi-modal models to refine rendered images, ensuring visually coherent results. Our strategy improves cross-view consistency without relying on the prompt provided for diffusion guidance, thereby bridging text-driven and image-driven methods.

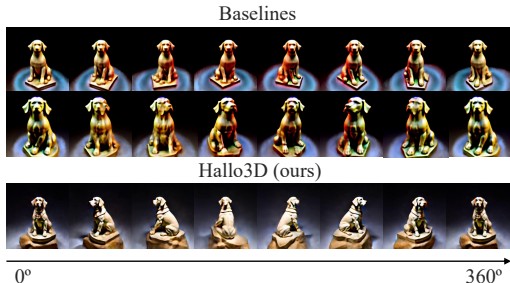

Figure 2: **Illustration of the Janus problem.** The first two rows show overfitting with repeated frontal views, while the third row, using Hallo3D, achieves more consistent results. This highlights the issue and clarifies the expected outcome.

In light of the findings above, we present **Hallo3D**, a novel, tuning-free approach that significantly enhances the multi-view consistency of 3D generation and is applicable across various generation methods. Our approach comprises three core techniques. Multi-Modal Hallucination Detection: This technique detects concrete hallucinations in renderings by leveraging large multi-modal models to represent inconsistency query information. Prompt-Enhanced Reconsistency: Utilizing the detection results from Multi-Modal Hallucination Detection as an enhanced negative prompt to precisely eliminate inconsistent artifacts in the renderings. Multi-view Appearance Alignment: This technique ensures uniform color and texture in renderings from different viewpoints by controlling attention in the diffusion de-noising process. By integrating these techniques, Hallo3D effectively addresses the challenges of maintaining consistency in 3D generation, providing a robust solution applicable to various generative methods. Experimental results demonstrate that our method exhibits a significant advantage in multi-view consistency compared to baseline models and can be robustly applied to various 3D generation tasks.

Our contributions can be summarized as follows:

- We propose Hallo3D, a novel tuning-free method that significantly enhances the multi-view consistency of 3D content generation and can be widely applied across various 3D generation paradigms, achieving outstanding experimental results.

- We demonstrate that large multi-modal models, unconstrained by geometry, can infer geometric structures and be utilized to detect and mitigate hallucinations in 3D generation.

- We introduce an optimization strategy that aligns the structures and surfaces of 3D content across views and addresses artifacts and hallucination through enhanced prompts.

## 2   Related Work

**Text-to-3D Generation.** The evolution of diffusion models has markedly enhanced text-to-3D generation. DreamFusion [40] initiated text-guided 3D modeling by using visual priors from 2D diffusion models to train 3D architectures, incorporating MipNeRF 360 [3] and Imagen [47]. While NeRF-based methods [33, 32, 48, 66, 59, 22, 12, 24, 10] handle complex lighting well, they are slower due to continuous parameter updates. In contrast, 3D Gaussian Splatting (3DGS) methods [20, 56, 5, 63, 39] have improved rendering speeds, showing their efficacy in complex scenarios.

**Image-to-3D Generation.** Images from specific views of a 3D model demonstrate improved visual consistency for 3D generation tasks. Image-based methods [53, 11, 30, 54, 1, 41] typically surpass text-based approaches by leveraging viewpoint-specific ground truth. 3D-aware image generation techniques [61, 7] utilize neural networks to enhance rendering beyond the primary viewpoint, although training data scarcity [23, 21] remains a challenge. Recent strides in integrating 3D visual data into 2D diffusion models [19, 36] have notably enhanced image-based 3D generation, reducing perceptual errors and improving generative quality.

**Methods for Enhancing Multi-view Consistency.** The primary challenge in enhancing 3D generation consistency is addressing hallucinations from 2D pre-trained diffusion models. A common strategy involves integrating additional 3D information into the diffusion process via fine-tuning [50, 65, 62, 29, 18]. This includes training models to handle consistent 3D subjects [46, 44], transparent backgrounds [64], and diverse viewpoints [49]. Recent advancements have also adjusted prompt embeddings to enhance viewpoint accuracy [17, 2], although this remains limited for non-orthogonal views. Alternatives include using geometric methods [25] or treating 3D generation analogously to video generation [57], though these tend to be framework-specific. In contrast, our method effectively optimizes arbitrary viewpoints, making it versatile across different 3D frameworks.

## 3   Methodology

### 3.1   Preliminaries

**Diffusion Models.** The diffusion model [15] has a fozrward diffusion process with diffusion steps from 0 to $T$, which degrades the original sample $\mathbf{x}_0$ into pure noise $\mathbf{x}_T$,

$$\mathbf{x}_t = \sqrt{\alpha_\mathbf{t}}\mathbf{x}_0 + \sqrt{1 - \alpha_\mathbf{t}}\boldsymbol{\epsilon}, \quad \boldsymbol{\epsilon} \sim \mathcal{N}(\mathbf{0}, \mathbf{I}), \tag{1}$$

where $t$ is the noise injection level, and $\boldsymbol{\alpha} := (\alpha_1, \ldots, \alpha_T) \in \mathbb{R}_{\geq 0}^T$ are hyper-parameters to determine noise scales at $T$ diffusion steps, and the reverse diffusion process is used during inference to generate $\mathbf{x}_0$ from $\mathbf{x}_t$. In text-guided diffusion models [52], the model is conditioned on text prompts $P$, which are converted into text embeddings via a text encoder such as CLIP [43]. The diffusion model $\boldsymbol{\epsilon}_\phi$ is trained using the MSE loss between the predicted noise $\hat{\boldsymbol{\epsilon}}_\phi$ and the actual noise $\boldsymbol{\epsilon}$,

$$L(\phi) = \mathbb{E}_{t \sim U(1,T), \boldsymbol{\epsilon} \sim \mathcal{N}(\mathbf{0}, \boldsymbol{I})} \|\boldsymbol{\epsilon} - \boldsymbol{\epsilon}_\phi(\mathbf{x}_t, t, P)\|_2^2,$$

where $U(1, T)$ represents a uniform distribution over the set $\{1, \cdots, T\}$, and $\mathcal{N}(\boldsymbol{\mu}, \boldsymbol{\Sigma})$ represents a multivariate Gaussian distribution with mean $\boldsymbol{\mu}$ and covariance $\boldsymbol{\Sigma}$. To enhance the alignment between text and images, Classifier-free guidance (CFG) [16] guides the generation of samples using

$$\hat{\boldsymbol{\epsilon}}_\phi(\mathbf{x}_t, t, P, \emptyset) = \boldsymbol{\epsilon}_\phi(\mathbf{x}_t, t, \emptyset) + s\left(\boldsymbol{\epsilon}_\phi(\mathbf{x}_t, t, P) - \boldsymbol{\epsilon}_\phi(\mathbf{x}_t, t, \emptyset)\right), \tag{2}$$

where $\emptyset$ is a special null text prompt representing the unconditional case, and $s > 0$ is the guidance scale. Increasing the guidance scale improves text-image alignment but reduces diversity. In practice, the $\emptyset$ text prompt is replaced with a negative prompt $P^-$ consisting of negative descriptions [9] to avoid undesired content in the generated samples.

**Score Distillation Sampling.** Score Distillation Sampling (SDS) employs vision priors from pre-trained 2D diffusion models to supervise 3D models, establishing it as a foundational learning method in the domain of 3D generation, initially proposed by DreamFusion [40]. A 3D representation model, parameterized by $\theta$, and the pre-trained diffusion model $\boldsymbol{\epsilon}_\phi$, together enable the rendering of an image $\mathbf{x} = g(\theta, \mathbf{c})$ from the 3D content, where $g(\cdot)$ is a differentiable generator to render $\mathbf{x}$ and $\mathbf{c}$ is the camera pose. To ensure that $\mathbf{x}$ consistently exhibits high quality from any view and to align the probability of $\mathbf{x}$ with $p(\phi)$, SDS introduces a score estimation function $\hat{\boldsymbol{\epsilon}}_\phi(\mathbf{x}_t; t, P)$. This function

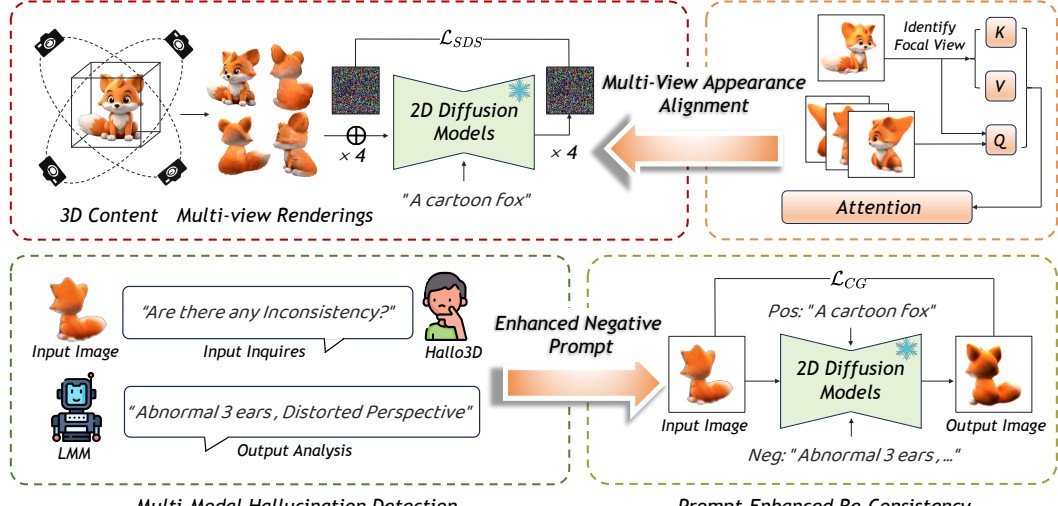

Figure 3: **Illustration of our pipeline.** We jointly optimize our model using $\mathcal{L}_{\text{SDS}}$ and $\mathcal{L}_{\text{CG}}$. For $\mathcal{L}_{\text{SDS}}$, we identify a focal view from multi-view renderings based on the camera pose, utilizing it as the keys (K) and values (V) to align all the four images using attention. This process harmonizes the appearance and feeds the output into the 2D Diffusion on the left, which plays a crucial role in refining the noise prediction. For $\mathcal{L}_{\text{CG}}$, we query hallucinations and inconsistencies in the rendering using an LMM and apply the results, outputted as enhanced negative prompt, to the following image optimization process to re-consistent a high-quality image. We calculate the $\mathcal{L}_{\text{CG}}$ based on the differences between the two images, thereby enhancing the consistency of the 3D content.

predicts the noise $\hat{\epsilon}_\phi$ based on the text condition $P$ and the noisy image $\mathbf{x}_t$, into which Gaussian noise $\epsilon$ has been injected. Furthermore, by calculating the discrepancy between $\hat{\epsilon}_\phi$ and $\epsilon$, the score function identifies the gradient direction for updating parameter $\theta$, thereby enhancing the training of the 3D model. The specific computation of the gradient is as follows:

$$\nabla_\theta \mathcal{L}_{\text{SDS}}(\phi, \mathbf{x} = g(\theta)) \triangleq \mathbb{E}_{t,\epsilon} \left[ w(t) \left( \hat{\epsilon}_\phi(\mathbf{x}_t; P, t) - \epsilon \right) \frac{\partial \mathbf{x}}{\partial \theta} \right], \quad (3)$$

where $w(t)$ is a weighting function.

## 3.2 Multi-view Appearance Alignment

Building upon our understanding of the SDS as discussed in Sec. 3.1, we further explored the impact of SDS on the consistency of appearances in 3D generation. We observed that SDS processes images from only one view at a time, which contradicts our intuition that enhancing 3D multi-view consistency should involve the simultaneous processing of multiple viewpoint images. Experimental results demonstrate that this approach led to a lack of interaction between images from different views during the training process, resulting in the loss of some surface information during the noise prediction process, as shown in Fig. 7 in the ablation study.

To circumvent the limitations of SDS, which typically favors image generation from a single view, we propose a "Multi-view Appearance Alignment" strategy. This approach introduces a consistent denoising method $\tilde{\epsilon}_\phi(\cdot)$ that incorporates an attention mechanism AAttn$(\cdot)$, enabling the rendering of multiple images from random views and providing a broader perspective compared to techniques focused primarily on single-view image generation [40, 58].

Specifically, inspired by recent advancements in diffusion models [27, 14, 4, 55, 35], which suggest that query features within attention spaces primarily shape image structure and layout, while key and value features influence texture, our method leverages this insight. As illustrated in the top right corner of Fig. 3, we select a focal view $i$ based on the camera pose, using the image from this viewpoint to provide the key and value features in the attention module. These are used to compute query features across all views, ensuring alignment of appearances. The attention is defined by the

Figure 4: **A multi-modal case study** for evaluating the capabilities of LMMs in 3D generation tasks. The first round of dialogue demonstrates that LMMs can infer structural consistency from 3D rendered images, while the second round shows that LMMs can respond in specific formats, allowing us to subsequently identify the negative prompts output using regular expressions.

following formula:

$$\text{AAttn}(Q, K_i, V_i) = \text{Softmax}\left(\frac{QK_i^T}{\sqrt{d}}\right) \cdot V_i, \tag{4}$$

where $\text{AAttn}(\cdot)$ is the appearance attention, with $K_i$ and $V_i$ as the key and value features corresponding to the image rendered from the focal view $i$, and $Q$ as the query feature from all views. The key and value are derived from the focal view, while each of the four views calculates a distinct query. In the denoising strategy $\tilde{\epsilon}_\phi(\cdot)$, this attention mechanism functions as cross-attention, aligning features from all views with the focal view to ensure consistent appearances. This process occurs within the U-Net network [45] in $\tilde{\epsilon}_\phi(\cdot)$, prior to the cross attention with the prompt.

## 3.3 Multi-modal Hallucination Detection

As shown in Fig. 4, the rendering image in the top right corner of the figure exhibits significant inconsistencies, due to the limitations of 2D pre-trained models in comprehending spatial concepts. This often leads to hallucinations and overfitting to specific viewpoints. However, we believe that Large Multi-modal Models (LMMs) have the capability to reason about and mitigate these hallucinations. To demonstrate this, we designed a two-phase inquiry involving LMMs, specifically using high-performing GPT-4V [38] and LLaVA [26] as examples. The dialogue depicted in the figure indicates that although LMMs were not explicitly trained with geometric constraints, they could identify inconsistencies in the 3D renderings and categorize them as negative prompts. Additionally, LMMs can standardize their output format based on a one-shot reference, making it easier for us to extract negative prompts.

Specifically, in our model, we input one 2D rendered image alongside 3D-aware inquiry prompts, denoted as $P_I$, into the multi-modal large modal to assist in automatically identifying inconsistencies present during the 3D generation process. To further mitigate hallucinations and correct inconsistencies, we have standardized the output format of the LMM, enabling it to accurately generate negative prompts based on the provided shots. These negative prompts can then be used to rectify distorted images in subsequent steps. Given their effectiveness in purposefully addressing inconsistencies, we refer to them as "Enhanced Negative Prompts". We formalize this process as follows:

$$P_E^- = \boldsymbol{D}_\psi(\mathbf{x}, P_I), \tag{5}$$

where $\boldsymbol{D}_\psi$ is the LMM parameterized by $\psi$, and $P_E^-$ is the enhanced negative prompt.

## 3.4 Prompt-Enhanced Re-consistency

With the enhanced negative prompt $P_E^-$ introduced in Sec. 3.3, a straightforward method to refine 2D renderings involves employing image editing algorithms to address inconsistencies. However,

existing approaches predominantly focus on adjustments to the null prompt [34] or the modification of positive prompts [14], which are generally ineffectual for altering the geometric structures in 2D images derived from 3D models. To address this limitation, we introduce a novel module for achieving re-consistency in 2D renderings, termed "Prompt-Enhanced Re-consistency," which leverages $P_E^-$ to effectively refine the geometric fidelity of the rendered images.

We regenerate the 2D rendered image $\mathbf{x}_0$ under the guidance of $P_E^-$. Specifically, to preserve the original semantic information of $\mathbf{x}_0$, we employ Denoising Diffusion Implicit Models (DDIM) [51] to invert theimage $\mathbf{x}_0$ to its noisy representation $\mathbf{x}_T$. Subsequently, we apply DDIM sampling to generate the consistent versions of the image, denoted as $\hat{\mathbf{x}}_0$, from $\mathbf{x}_T$ as follows:

$$\hat{\mathbf{x}}_{t-1} = \sqrt{\frac{\alpha_{t-1}}{\alpha_t}}\hat{\mathbf{x}}_t + (\sqrt{1-\alpha_{t-1}} - \sqrt{\frac{\alpha_{t-1}(1-\alpha_t)}{\alpha_t}})\tilde{\epsilon}_\phi(\hat{\mathbf{x}}_t, t, P^+, P_E^-) \qquad (6)$$

where $\tilde{\epsilon}_\phi(\hat{\mathbf{x}}_t, t, P^+, P_E^-)$ is the denoising strategy incorporated the attention mechanism $\text{Attn}(\cdot)$ in Sec. 3.2, and is adjusted by Classifier-Free Guidance (CFG) [16], with the null text prompt replaced by the enhanced negative prompt $P_E^-$. This approach ensures that the regenerated image retains its core semantic integrity while improving its multi-view consistency. After completing $T$ iterations as delineated by Eq. 1, we successfully achieve the re-consistent image $\hat{\mathbf{x}}_0$, effectively reconciling the image consistency with its original semantic information.

Finally, we train the 3D model $\theta$ using the MSE loss $\mathcal{L}_{\text{CG}}$ between $\mathbf{x}_0$ and $\hat{\mathbf{x}}_0$ in the image space:

$$\mathcal{L}_{\text{CG}} \triangleq \mathbb{E}\left[(\hat{\mathbf{x}}_0 - \mathbf{x}_0)\right], \qquad (7)$$

It is worth noting that we apply Prompt-Enhanced Reconsistency only when the rendered image exhibit complete semantic structure. Our detector, $\boldsymbol{D}_\psi$, assesses the semantic completeness of the image. If the semantic structure is deemed incomplete or unclear, $\boldsymbol{D}_\psi$ returns None, precluding further processing. This ensures that enhancements are only applied to images that are adequately prepared. The dependency of our enhancement process on the state of semantic completeness directly influences the formulation of the final training loss for the 3D model, as detailed below:

$$\mathcal{L}(\theta) = \begin{cases} \mathcal{L}_{\text{SDS}} + w\mathcal{L}_{\text{CG}}, & \text{if } \boldsymbol{D}_\psi(\mathbf{x}, P_I) \text{ is not None}, \\ \mathcal{L}_{\text{SDS}}, & \text{otherwise}. \end{cases} \qquad (8)$$

where $w$ is set to balance the magnitude of $\mathcal{L}_{\text{SDS}}$ and $\mathcal{L}_{\text{CG}}$. By incorporating $\mathcal{L}_{CG}$, which is only applied when $D_\psi$ confirms the semantic readiness of the image, we ensure that our model focuses on enhancing well-formed images. This selective application of $\mathcal{L}_{CG}$ prevents further exacerbating the quality of images already of poor quality. Simultaneously, it avoids misallocating resources to images that do not benefit from the intended enhancements, thereby improving the efficiency and effectiveness of our training process. For more implementation details, see the Appendix.A.

## 4  Expremients

In this section, we comprehensively evaluate Hallo3D's performance within two categories of 3D generation frameworks: text-to-3D and image-to-3D. We present comparative results with other baseline models to highlight its capabilities. Additionally, to further substantiate the effectiveness of Hallo3D in enhancing multi-view consistency in 3D generation, we have conducted an extensive user study. Finally, we designed ablation experiments to validate the necessity of the framework's design.

### 4.1  Experiment Setup

**Baselines.** We evaluated our method against several established baselines, demonstrating strong performance across diverse frameworks. These include text-to-3D models like GaussianDreamer [63], Score Jacobian Chain (SJC) [58], DreamFusion-IF [40], and Magic3D [24], as well as image-to-3D models such as DreamGaussian [53] and Zero-1-to-3 [28], an extension of DreamFusion. We also included methods based on NeRF [33] and 3DGS [20] for a comprehensive comparison. Identical parameter configurations and seed values were maintained for fair comparison, using default hyperparameters from the baselines' open-source implementations. We employed the Threestudio library [13] for SJC and Magic3D, and the official codebases for the other methods. Additionally, we conducted experiments to evaluate the time consumption of Hallo3D, detailed in the Appendix.B.

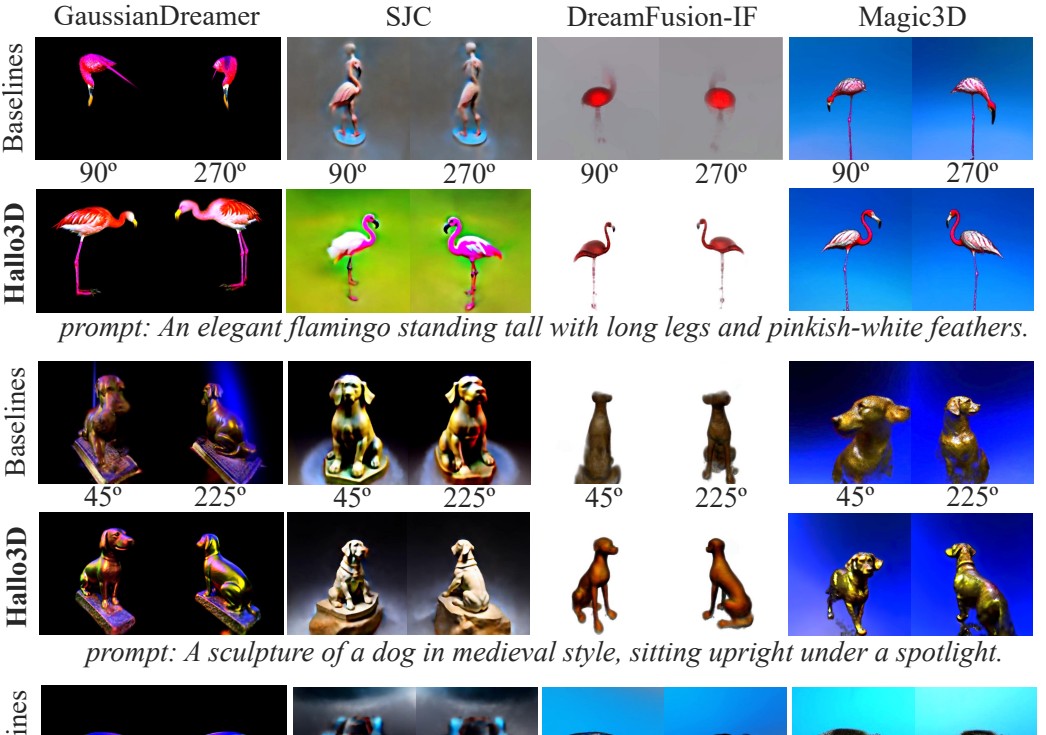

Figure 5: **Qualitative comparison in text-driven 3D generation** of HalloD and baseline models. To provide a more straightforward comparison, we rendered both Hallo3D and the baseline models from two identical and complementary angles.

Table 1: Quantitative comparisons in text-driven 3D generation

| Metrics | GaussianDreamer | Hallo3D | SJC | Hallo3D | DreamFusion-IF | Hallo3D | Magic3D | Hallo3D |
|---|---|---|---|---|---|---|---|---|
| **CLIP-Score B/32** ↑ | 21.31 | **24.53** | 20.13 | **24.34** | 14.09 | **22.15** | 14.93 | **22.05** |
| **CLIP-Score B/16** ↑ | 22.67 | **27.00** | 21.36 | **26.36** | 15.98 | **23.79** | 16.41 | **24.29** |
| **CLIP-Score L/14** ↑ | 23.70 | **30.12** | 23.95 | **28.04** | 18.19 | **26.72** | 17.99 | **27.72** |

Table 2: User study in text-driven 3D generation

| Metrics | GaussianDreamer | Hallo3D | SJC | Hallo3D | DreamFusion-IF | Hallo3D | Magic3D | Hallo3D |
|---|---|---|---|---|---|---|---|---|
| **Multi-view Consistency** ↑ | 6.00 | **8.87** | 4.53 | **7.63** | 4.63 | **6.33** | 5.13 | **7.53** |
| **Overall Quality** ↑ | 5.53 | **8.67** | 4.77 | **7.80** | 4.17 | **7.37** | 4.60 | **8.03** |
| **Alignment with Prompt** ↑ | 5.57 | **8.87** | 5.63 | **7.40** | 4.70 | **7.03** | 5.17 | **7.37** |

**Metrics.** The field of 3D generation struggles with the absence of ground truth, complicating the development of a unified evaluation metric. To address multi-view consistency, we reviewed existing evaluation methods and identified 3D inconsistencies using CLIP-Score [43]. We generated 80 unique 3D prompts using ChatGPT [37] and arranged 16 cameras in a 360-degree configuration around the z-axis. The average CLIP-Score across all views measured consistency.

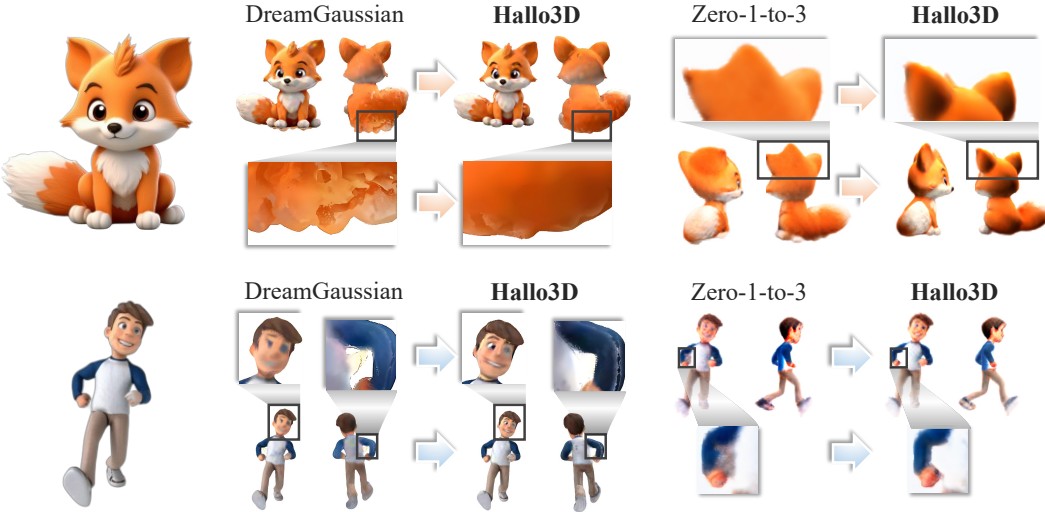

Figure 6: **Qualitative comparison in image-driven 3D generation** of Hallo3D and baseline models. To facilitate a more direct comparison, we rendered both Hallo3D and the baseline models from two complementary angles and magnified specific details.

Table 3: User study in image-driven 3D generation

| Metrics | DreamGaussian | **Hallo3D** | Zero-1-to-3 | **Hallo3D** |
|---|---|---|---|---|
| **Multi-view Consistency** ↑ | 8.40 | **9.15** | 7.25 | **7.81** |
| **Overall Quality** ↑ | 9.23 | **9.52** | 6.10 | **7.20** |
| **Alignment with Prompt** ↑ | 8.55 | **8.00** | 8.30 | **8.95** |

## 4.2 Quantitative Comparison with Baselines

In our qualitative evaluation for text-driven 3D content generation, we randomly selected three prompts from a dataset of 80 and used two high-definition images from Google Images for image-driven 3D generation. The results, shown in Fig. 5 and Fig. 6, reveal significant enhancements in multi-view consistency. Baseline models often produced flawed figures, such as headless "flamingos" or "dogs" with multiple heads and ears. In contrast, our models achieved more realistic and consistent outputs, confirming the effectiveness of our approach. The 360-degree visualization is shown in Appendix.C.

## 4.3 Qualitative Comparison with Baselines

**Computational results.** Following [63, 42, 53], we evaluated the CLIP-Score to assess the quality and consistency of 3D generated contents, as presented in Tab.1. The results indicate that our approach outperforms all baseline models, confirming the effectiveness of our method. It should be noted that the existence of a ground truth corresponding to the front view in image-driven 3D generation generally leads to higher generation quality.

Consequently, for image-3D tasks, we adhered to the experimental setup outlined in [31, 60]. Specifically, we selected 60 objects from the GSO [8] and Objaverse [6] datasets, replacing overly simple objects to ensure a more robust evaluation. These objects were rendered in frontal views at a resolution of 256x256. To comprehensively assess performance, we utilized Chamfer Distance (CD) and Volume IoU (Vol. IoU) for evaluating geometric accuracy, along with PSNR, SSIM, and LPIPS for measuring visual quality. As presented in Tab.4, the experimental results clearly indicate that our method surpasses the baseline across all metrics, achieving significant improvements in both geometry and texture quality. This further substantiates the broad applicability of our approach, demonstrating its capacity to enhance both text-to-3D and image-to-3D tasks.

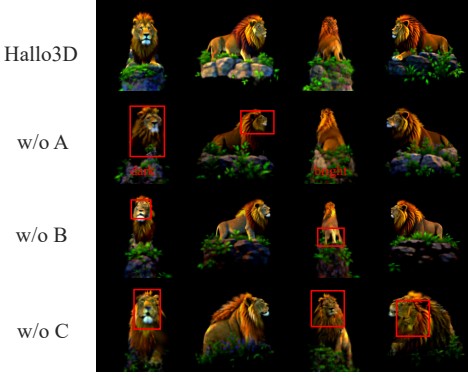 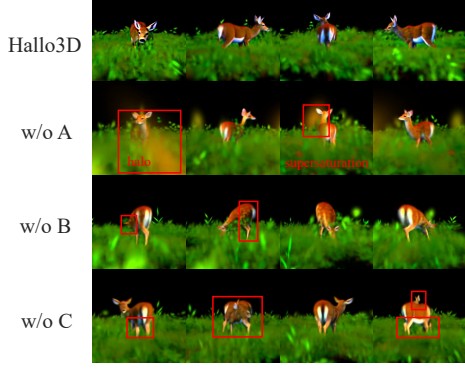

*prompt: A majestic lion standing on a rock.*  *prompt: A deer grazing on a grassy plain.*

Figure 7: **Ablation study of our method.** In the figure, module A represents Multi-view Appearance Alignment in Sec. 3.2, module B stands for Multi-modal Hallucination Detection in Sec. 3.3, and module C denotes Prompt-Enhanced Re-Consistency in Sec. 3.4. We conducted ablation studies on each of these three modules respectively.

Table 4: Quantitative comparisons in image-driven 3D generation

| Metrics | DreamGaussian | **Hallo3D** | Zero-1-to-3 | **Hallo3D** |
|---------|---------------|-------------|-------------|-------------|
| **CD↓** | 0.0185 | **0.0171** | 0.0370 | **0.0283** |
| **Vol. IoU↑** | 0.5861 | **0.6099** | 0.4824 | **0.5602** |
| **PSNR↑** | 16.502 | **16.518** | 13.433 | **14.930** |
| **SSIM↑** | 0.8543 | **0.8793** | 0.7210 | **0.7527** |
| **LPIPS↓** | 0.2025 | **0.1726** | 0.3926 | **0.3328** |

**User study.** In our user study, we recruited 58 volunteers with expertise in artificial intelligence to participate in our experiment. To comprehensively assess the quality discrepancies among various generated 3D models, we developed an extensive scale for the volunteers to fill out. Specifically, we generated 120-frame videos for each 3D model, totaling 32 video sets. Our comparative approach involved evaluating each model independently on three criteria: "Multi-view Consistency," "Overall Quality," and "Alignment with Prompt," with ratings on a scale from 1 to 10. The findings were summarized by compiling the average scores, and can be seen in Tab.2 and Tab.3.

### 4.4 Ablation Study

We conducted ablation experiments on the three Hallo3D modules, as shown in Fig. 7. Starting from the complete model, we independently removed each module and assessed their effects. Notably, in the "w/o C" setting, the output of LMM, $P_E^-$ is applied to $\mathcal{L}_{SDS}$ calculations to demonstrate the necessity of $\mathcal{L}_{CG}$. Additionally, we conducted an ablation on "w/o C & $P_E^-$", where $P_E^-$ is not applied anywhere in the "w/o C" setting, to further highlight the effectiveness of the module.

In Fig.7, we focus on how module A primarily affects color and texture, while module B and module C enhance cross-view consistency. Specifically,

- In Row 2, w/o A: The lion appears significantly darker than in Row 1, and the deer exhibits a blurry halo accompanied by an unnatural color shift.

- In Row 3, w/o B: The lion's head is noticeably deformed in both the first and third columns, while the deer entirely loses its head.

- Row 4, w/o C: The "second face" appears on the lion's back in the third column, and the right deer's image shows a clear Janus Problem, with multiple legs and a distorted body visible in the second and fourth columns.

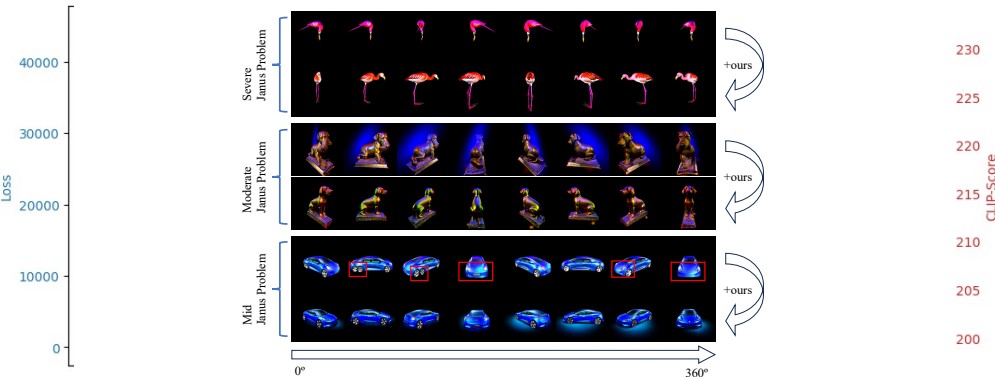

Figure 8: Loss curves for $\mathcal{L}_{\mathrm{CG}}$ and $\mathcal{L}_{\mathrm{SDS}}$, along with the CLIP-Score curves with and without $\mathcal{L}_{\mathrm{CG}}$.

Table 5: Quantitative Ablation Results

| Metrics | Hallo3D | w/o A | w/o B | w/o C | w/o C & $P_E^-$ | Baseline |
|---|---|---|---|---|---|---|
| CLIP-Score B/32↑ | 24.25 | 23.98 | 23.65 | 22.46 | 22.23 | 21.27 |
| CLIP-Score B/16↑ | 26.83 | 25.88 | 25.10 | 23.59 | 23.23 | 22.67 |
| CLIP-Score L/14↑ | 30.00 | 29.36 | 28.71 | 26.92 | 25.58 | 23.71 |

Additionally, we conducted quantifiable ablation experiments, as shown in Tab.4.4, further demonstrating the effectiveness and necessity of each module in Hallo3D. Moreover, the better performance of "w/o C" compared to "w/o C & $P_E^-$" also supports the necessity of introducing $\mathcal{L}_{\mathrm{CG}}$.

## 4.5 Balance between $\mathcal{L}_{\mathrm{CG}}$ and $\mathcal{L}_{\mathrm{SDS}}$.

In our experiments, we observed that the loss function $\mathcal{L}_{\mathrm{CG}}$, which is computed on a per-pixel basis, typically exhibits a larger magnitude in comparison to $\mathcal{L}_{\mathrm{SDS}}$. To achieve a balanced scale between these losses, we assigned a weight of $w = 0.1$ to $\mathcal{L}_{\mathrm{CG}}$ in Eq.8. It is important to note that this adjustment in weight does not diminish the importance of $\mathcal{L}_{\mathrm{CG}}$ in any way. Specifically, as shown in Fig.8, even after scaling $\mathcal{L}_{\mathrm{CG}}$ by the factor $w$, it maintains a larger magnitude compared to $\mathcal{L}_{\mathrm{SDS}}$. This demonstrates that $\mathcal{L}_{\mathrm{CG}}$ provides ample guidance for 3D generation, ensuring effective optimization throughout the process.

## 5 Conclusion

In this paper, we introduce Hallo3D, a novel approach designed to enhance 3D content generation through both text-driven and image-driven methods. We demonstrate the capability of large multi-modal models to infer geometric structures and detect hallucination arising from 2D diffusion models. By combining LMM with diffusion models, we achieve re-consistent 2D images applicable in the 3D domain. Extensive experimental evidence substantiates that our method significantly improves consistency and mitigates hallucinations in 3D content generation. Additionally, we thank Haoyang Tong from the University of Chinese Academy of Sciences for his contributions to this work.

## Acknowledgements

This research is supported in part by several funding sources, including the National Natural Science Foundation of China (Grant No. 32341009, 62206277, U21B2045, 62425606). Additional support has been provided by the Beijing Nova Program (20230484276), as well as the Youth Innovation Promotion Association of the Chinese Academy of Sciences (Grant No. 2022132). These contributions have been instrumental in enabling the advancement of this work.

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

# A  Implementation details

**Selection of focal view.** We use Fovy, the camera's vertical view, as our selection standard. The first view with Fovy exceeding 120% of each baseline's default becomes our focal view, enabling a broader shot capturing more object details.

**Details of LMM setup.** We used different $P_I$ than in Fig.4. The primary purpose of Fig.4 is to use a case study to demonstrate how LMMs can infer structural consistency and respond in specific formats. To highlight this capability, we employed two dialogues. In practice, we used a single interaction to query the LMM to achieve faster runtime. The specific setting is as follows.

> "You are a master of 3D generation, and please refer to the 'Prompt' and 'Negative Prompt' below to identify the inconsistency in the image I provided you with, with body shape, perspective, texture, and so on.
> Reference:
> 'Prompt': '3d render of xx, front view, standing, high quality, 4K',
> 'Negative Prompt': 'multi-head, unnatural lighting, smooth appearance, distorted color, long neck, two-nosed, extra limbs' ".

For LMM, we chose the locally deployed LLaVA [26], using the version llava-v1.6-34b.

**General prompt.** Our method acts as a universal enhancement for 3D generation, considering the common use of general negative prompts in baseline methods [63, 58, 40, 53, 24, 28]. The specific setting is as follows.

> "unnatural colors, poor lighting, low quality, artifacts, smooth texture".

# B  Time Consumption

We recorded the runtime using two baselines: GaussianDreamer [63] based on 3DGS with fewer iterations and faster speed, and DreamFusion [40] based on NeRF with more iterations and slower speed, on NVIDIA V100.

To optimize the process, we begin calculating $\mathcal{L}_{CG}$ later in the training and only every 4 iterations in our experiments. This approach is consistent with the statement in Sec.3.4 that *"this module only works when the rendered images exhibit complete semantic structures."* The rationale is twofold: first, during the early stages of training, the 3D assets are relatively disorganized and lack clear semantic structures, making it challenging for LMMs to reason accurately. Therefore, we delay the introduction of $\mathcal{L}_{CG}$. Second, we empirically found that calculating $\mathcal{L}_{CG}$ every 4 iterations does not affect performance, allowing us to reduce training time. The results are presented in the Tab.6.

Notably, since our method includes the "Multi-View Appearance Alignment" module, which requires attention calculations across four differently angled rendered images, we set the batch size to 4 for all baselines. To ensure a fair comparison, we reduced the number of iterations to 1/4 of the original. For example, DreamFusion originally trained for 10,000 iterations, and we adjusted it to 2,500 for optimization. GaussianDreamer(iteration=1200) already uses batch=4, so we matched its iteration count at 1,200.

The experimental results indicate that while our method introduces some additional time overhead, this is fully justified by the significant improvements in performance and quality, especially considering the challenging nature of addressing the Janus Problem.

# C  Additional Experiments

## C.1  The Effectiveness and Necessity of $\mathcal{L}_{CG}$

To further underscore the necessity of incorporating $\mathcal{L}_{CG}$, we plotted the curve of $\mathcal{L}_{CG}$ over the course of iterations. In conjunction with this, we also plotted the CLIP-Score for both the complete model and an ablated version that omits $\mathcal{L}_{CG}$. As illustrated in Fig.8, it is evident that $\mathcal{L}_{CG}$ steadily decreases with increasing iterations, contributing to a marked improvement in the CLIP-Score. In

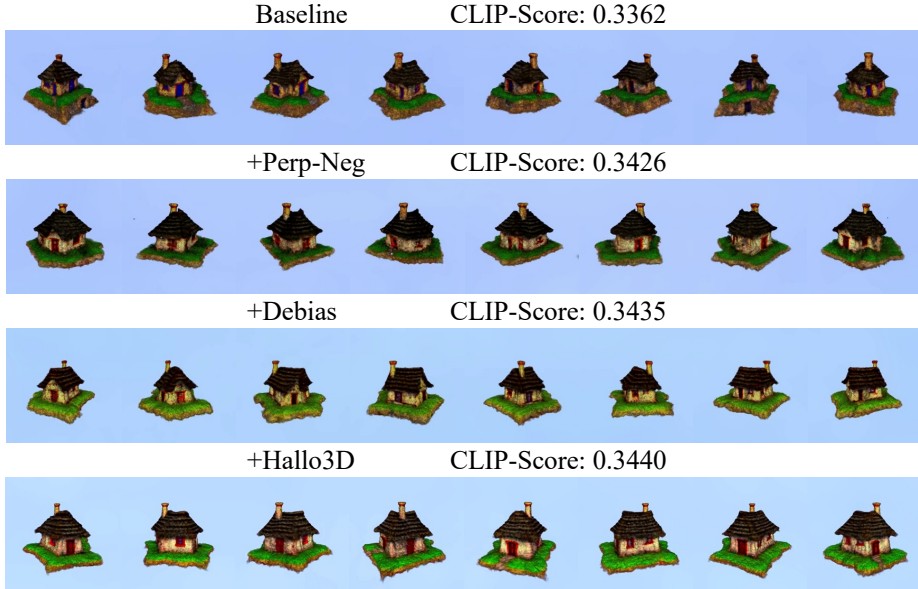

Baseline          CLIP-Score: 0.3362

+Perp-Neg         CLIP-Score: 0.3426

+Debias           CLIP-Score: 0.3435

+Hallo3D          CLIP-Score: 0.3440

*prompt: A 3D model of an adorable cottage with a thatched roof.*

Figure 9: Comparison experiments with Perp-Neg and Debias.

Table 6: The time consumption introduced by Hallo3D.

| Baseline | Iteration | $\mathcal{L}_{CG}$ Start Rounds | Original Time | Extra Time | Total Time |
|---|---|---|---|---|---|
| GaussianDreamer | 1200 | 1000 | ~28 min | ~10 min | ~38 min |
| DreamFusion | 2500 | 2200 | ~51 min | ~15 min | ~66 min |

contrast, the CLIP-Score for the model without $\mathcal{L}_{CG}$ exhibits only a marginal improvement. These findings clearly highlight both the necessity and effectiveness of incorporating $\mathcal{L}_{CG}$ into the model.

### C.2 Comparison Experiments with Other 3D Consistency Enhancement Methods.

To further demonstrate the advantages of our method, we compared it with other approaches[2, 17] aimed at improving 3D consistency. As shown in Fig.9, Hallo3D more effectively addresses the Janus problem and achieves a greater improvement in CLIP-Score.

### C.3 360-degree Visualization Results

Due to space constraints, Fig.5 in the main text only presents 3D generation results from selected viewpoints. The full 360-degree visualizations can be found in Fig.10 and Fig.11.

## D   Limitation

Our method has shown improvements in 3D generation consistency across various baselines, including both text-based and image-based approaches. However, as a method focused on enhancing view consistency, the quality of our experimental results is inherently tied to the performance of the baseline models. Moreover, the potential misuse of advanced 3D generation technologies poses risks to social trust and information integrity. Looking ahead, we will prioritize the Janus Problem as a key research direction and are committed to contributing further to the field of 3D generation alongside our fellow researchers.

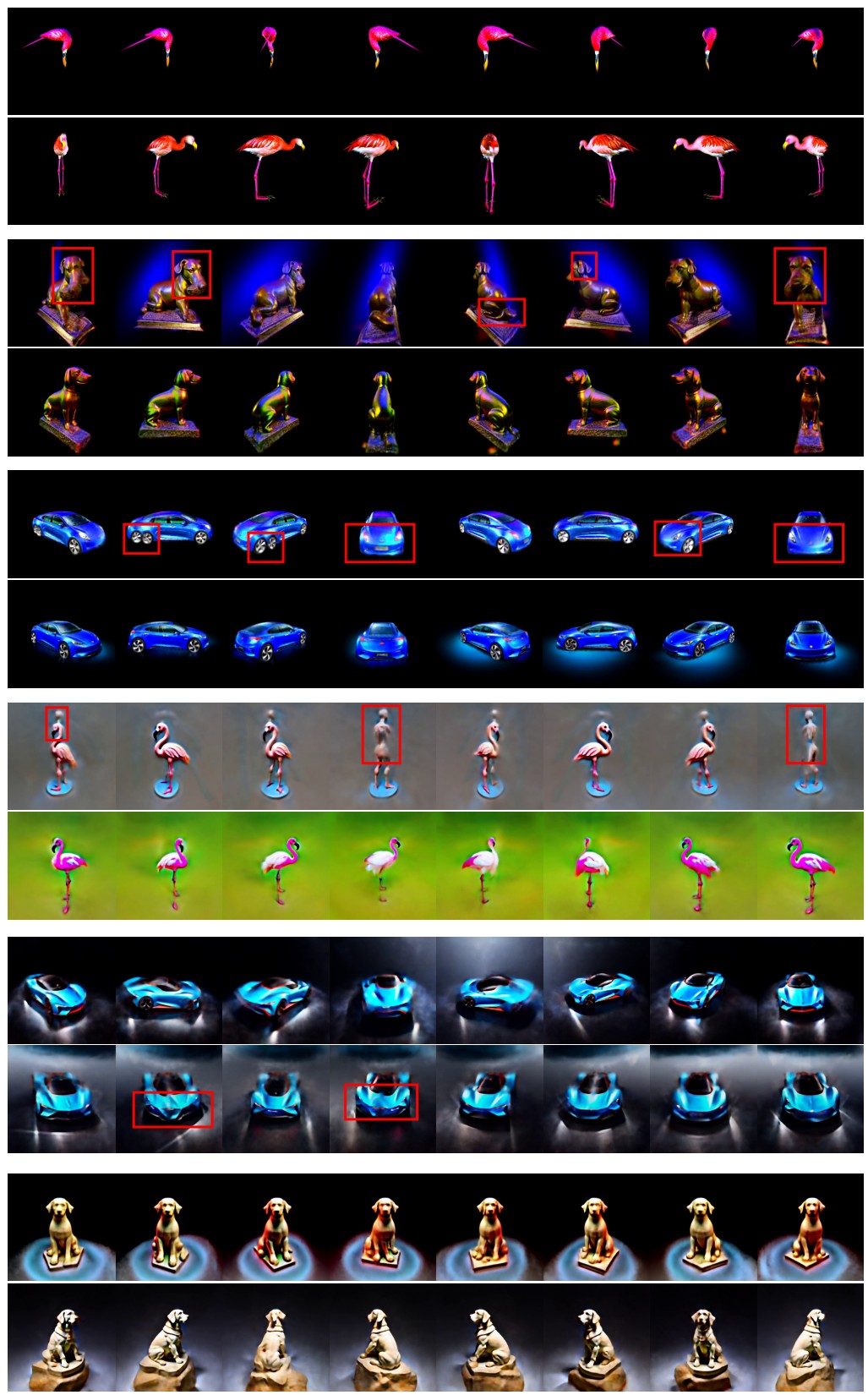

Figure 10: 360-degree visualization results in Fig.5 (1).

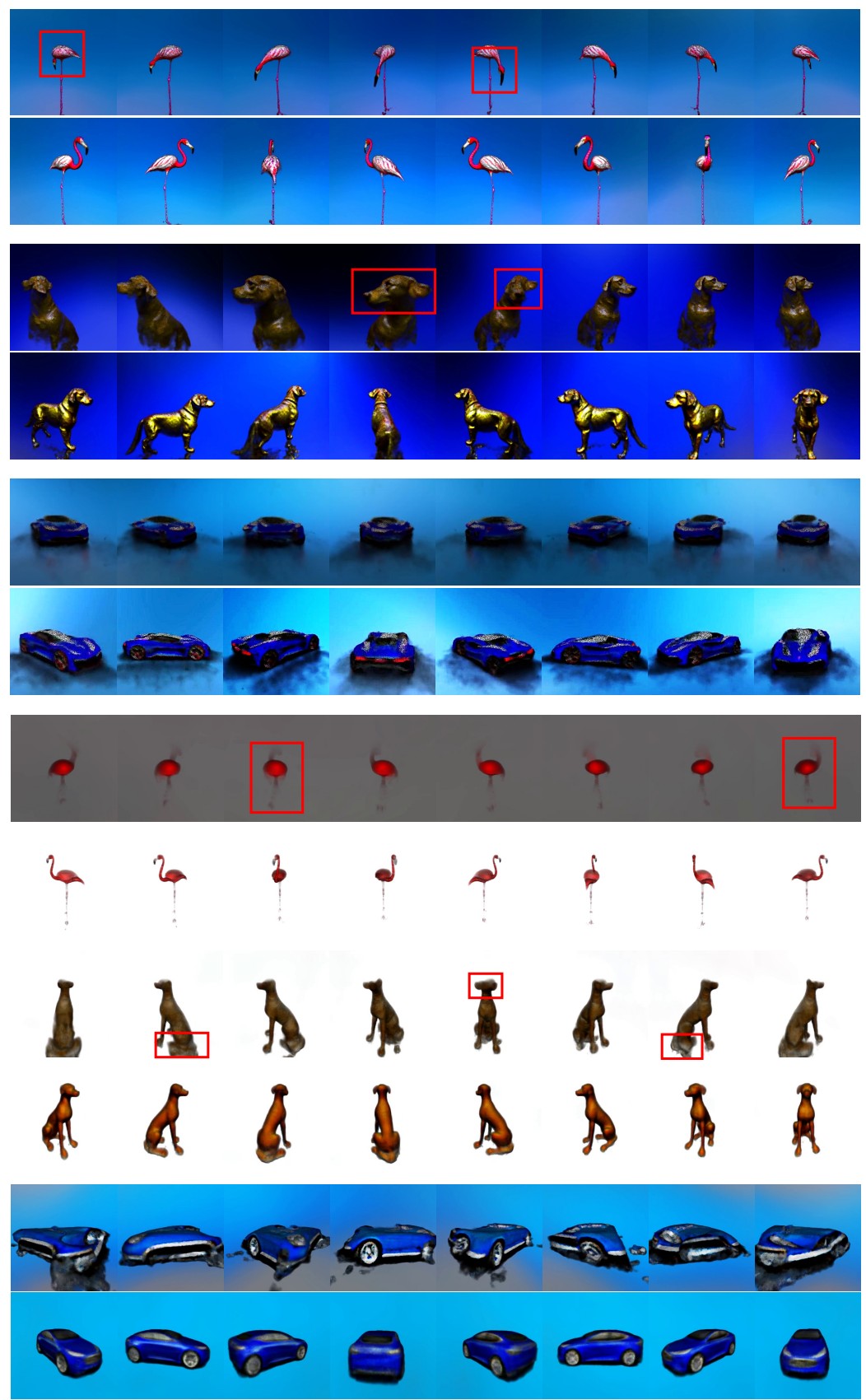

Figure 11: 360-degree visualization results in Fig.5 (2).

