# OpenReview forum: "Hallo3D: Multi-Modal Hallucination Detection and Mitigation for Consistent 3D Content Generation"
_NeurIPS.cc/2024/Conference — NeurIPS 2024 poster_

### Official Review · Reviewer_isRS · 2024-06-24

**Soundness:** 3
**Presentation:** 3
**Contribution:** 3
**Rating:** 6
**Confidence:** 4

**Summary:**

The authors point out that generating 3D content with Score Distillation Sampling leads to multi-view inconsistencies. To address this challenge, they propose a novel, tuning-free method called Hallo3D. Specifically, they utilize large multi-modal models (e.g., GPT-4V) to detect and correct these inconsistencies during optimization. Notably, their method can be implemented in a plug-and-play manner with several text-to-3D generation baselines, achieving satisfying visual results.

**Strengths:**

1.	The idea of using large multi-modal models to implement a generation-detection-correction paradigm for text-to-3D generation is both novel and interesting. Moreover, the experimental results demonstrate that this approach works quite well.

2.	The entire "Methodology" section is presented quite well. The preliminaries are written accurately, and the subsequent subsections clearly introduce the overall motivation and design.

3.	The experiments are also solidly conducted, including both qualitative and quantitative comparisons on text- and image-driven 3D generation. The ablation studies are sensible and include user studies as well.

**Weaknesses:**

The overall quality of the paper is quite good, but some problems still exist:

1.	In Figure 2, the "Illustration of the Janus problem" seems to understate the performance of current baselines. In the first two rows of the figure, eight faces of the dog are visible around the sculpture, which does not align with the experimental results from methods like DreamFusion. This is acceptable as an illustration, but it would be better to be more rigorous.

2.	Equation 6 in line 183 does not seem to be derivable from Equation 12 in the original DDIM paper. Please check this carefully.

3.	The ablation study section lacks clarity. Rows 2-4 do not show a clear visual difference, and the text in Section 4.4 does not provide an easy-to-follow illustration.

**Questions:**

1.	Section 3.4 is not clear enough. From lines 180-183, the authors seem to use the “DDIM Inversion” technique to regenerate an image without the so-called hallucination. However, generating such an image with CFG without damaging the original structure of the image is not an easy task (see Null-text Inversion). More illustration of this part should be provided.

2.	From Figures 3 and 4, it appears that you only feed a single image into the LMM for hallucination detection. However, a less serious multi-view inconsistency (e.g., a rabbit with three eyes) can only be observed from certain view directions. How can the effectiveness of Hallo3D be proven in this context?

3.	The generation-detection-correction paradigm introduced in this paper appears to require frequent use of the LMM. Given that generating 3D content with SDS optimization typically requires around 5,000-10,000 iterations, what is the overall time cost of your algorithm?

I’m willing to give this paper a higher rating if these three questions are well addressed!

**Limitations:**

Yes, the authors discuss the limitations of the proposed Hallo3D clearly at the end of the "Conclusion and Discussion" section.

---

> ### Author Rebuttal · Authors · 2024-08-07
>
> Thank you for your feedback. We appreciate your recognition of our method's innovation and effectiveness. Here are our responses to your comments.
>
> **W1. "Fig.2 seems to understate the performance of current baselines."**
>
> A1. We appreciate your attention to the details in Fig.2. Actually, this illustration is based on the Score Jacobian Chain[1] (a text-to-3D baseline method). The figure's primary purpose is to visually explain the Janus problem by highlighting an example where the multi-faces phenomenon is particularly pronounced. To better illustrate the Janus Problem, we included the expected result under normal conditions as achieved by our method, focusing on the problem's severity rather than comparing different methods. We understand the importance of precision and will consider adding a note in the figure caption to ensure this context is clearer in future versions.
>
> **W2. "Equation 6 in line 183 does not seem to be derivable from Equation 12 in the DDIM paper."**
>
> A2. Thank you for your thorough review and for pointing out this issue. After carefully revisiting our derivation, we confirmed that there is indeed a typo. We have corrected the derivation by setting $\sigma =0$ in Eq.12 of the DDIM paper. The correct form is as follows:
>
> $$
> \hat{\mathbf{x}} _ {t-1} = \sqrt{\frac{\alpha _ {t-1}} {\alpha _ t}} \hat{\mathbf{x}} _ t +  (\sqrt{1-\alpha _ {t-1}}-\sqrt{\frac{\alpha _ {t-1} (1-\alpha _ t)} {\alpha _ t}}) \tilde{\epsilon} _ {\phi} (\hat{\mathbf{x}} _ t, t, P^{+}, P^{-} _ E)
> $$
>
> Your feedback has been instrumental in ensuring the precision of the paper. We have revised the formula accordingly in the updated version to enhance its accuracy.
>
> **W3. "Ablation study section lacks clarity. Rows 2-4 do not show a clear visual difference."**
>
> A3. To demonstrate the necessity of each module in our method, we conducted quantitative ablation experiments, including one that ablates both C and $P _ E ^ -$, i.e., "w/o C & $P _ E ^ -$" to validate the importance of $\mathcal{L} _ {\rm{CG}}$. The results in Q1 of the Global Review confirm that each module in Hallo3D enhances performance.
>
> In Sec.4, we focus on how module A primarily affects color and texture, while module B and module C enhance cross-view consistency. Specifically:
>
> - Row 2, w/o A: The lion is darker than Row1, and the deer shows a blurry halo and unreasonable color shift.
> - Row 3, w/o B: The lion's head is derormed in the first and third columns, and the deer misses its head.
> - Row 4, w/o C: The "second face" appears on the lion's back in the third column, and the right deer's image shows a clear Janus Problem, with multiple legs and a distorted body visible in the second and fourth columns.
>
> **Q1. "The authors use the “DDIM Inversion” to regenerate an image without hallucination. However, generating such an image with CFG without damaging the original structure is not easy."**
>
> A-Q1. That's an excellent question. We agree with your point that CFG-based techniques often struggle to maintain structure in regenerated images in image editing or reconstruction. However, in generative tasks, we use DDIM Inversion to regenerate the image as a guidance for constraining consistency, not as the final output. Our primary focus is on the final 3D assets, with the images obtained using rendering techniques that are independent of the diffusion model.
>
> Specifically, generative tasks involve repeated optimization, where each iteration may cause some structural loss, but these intermediary images are not visible. Therefore, as long as the final result is consistent and high-quality, the loss of information is relatively less important. This contrasts with image editing or reconstruction, where only a single image is produced, making structural details critical.
>
> Our experiments confirm this. As shown in Figures 5 and 6, even with structural information loss during the DDIM Inversion process, we achieved consistent results.
>
> **Q2. "It appears that you only feed a single image into the LMM. However, a less serious inconsistency can only be observed from certain view directions. How can the effectiveness of Hallo3D be proven?"**
>
> A-Q2. Yes, we only input one image at a time to reduce the inference cost of LMM. However, since the camera pose for each rendered image is random, multiple iterations ensure that all sides of the 3D object are addressed, effectively avoiding inconsistencies in specific views.
>
> Our experiments also show that this method effectively addresses even minor multi-view inconsistencies. For instance, in the second image of the fifth row in Fig.5, the car's right rear wheel appears doubled, but our method corrected this. Similarly, in the second example in Fig.6, our method improves subtle Janus problems on the figure's body.
>
> **Q3. "The method appears to require frequent use of the LMM. Given that generating 3D requires 5000-10000 iterations, what‘s the time cost?"**
>
> A-Q3. We measured the time overhead for both 3DGS-based and NeRF-based baselines, finding minimal additional costs, as detailed in Q1 of the Global Review. While LMM inference can add time, our model uses single-stage dialogue and queries at intervals during later training to improve efficiency. Additionally, due to the Multi-View Appearance Alignment, our method processes four images at a time with batch=4, reducing the number of SDS iterations.
>
> ---
> We hope that our response addresses your concerns sincerely. Looking forward to further communication with you!
>
> [1] Wang H,et al. Score jacobian chaining: Lifting pretrained 2d diffusion models for 3d generation. In *CVPR*, 2023.

---

> > ### Comment · Reviewer_isRS · 2024-08-11
> > **comment**
> >
> > I think the authors have addressed most of my concerns, and proved the feasibility of their work. Thus I decide to raise my grade from 5 (borderline accept) to 6 (weak accept).

---

> > > ### Author Response · Authors · 2024-08-11
> > > **Official Comment by Authors**
> > >
> > > Thank you for recognizing our work! We’re pleased to have addressed most of your concerns, and we will incorporate your suggestions in future versions.

---

### Official Review · Reviewer_bBV6 · 2024-07-03

**Soundness:** 2
**Presentation:** 2
**Contribution:** 3
**Rating:** 5
**Confidence:** 4

**Summary:**

Hallo3D presents a tuning-free method, empowering 3D content generation frameworks via multi-modal LLM. The paper aims to solve the hallucination and inconsistent problems in SDS-based 3D content generation pipelines. With the design of multi-view appearance alignment and enhanced negative prompts, Hallo3D is able to improve the quality of generated 3D objects. The effectiveness of the proposed method was demonstrated through experimental verification using multiple frameworks.

**Strengths:**

1. The paper integrates various SDS-based 3D generation frameworks, including text-to-3D and image-to-3D frameworks. And the effectiveness of the proposed method on these frameworks has been demonstrated through qualitative results.
2. Introducing multi-modal LLM to optimize the generation of diffusion is effective, and the output of multi-modal LLM can be directly applied to the diffusion through negative prompts. This design is concise and clever.
3. The paper also performs some quantitative results to validate the effectiveness of the proposed method.
4. The design is flexible and adaptable to the rapid advancements in SDS-based methods.

**Weaknesses:**

1. The method description is not clear enough.
- In lines 148-150, the paper claims defining a new denoising strategy, aiming to enhance the appearance consistency. But there's no further details about how to apply the new denoising strategy with appearance attention.
- Lines 184-186 provide some description, but lack a detailed explanation of how to utilize appearance attention, which is an important contribution claimed by the paper.
2. The ablation study lacks quantitative analysis.
- Figure 7 presents qualitative results; however, the lack of quantitative analysis makes these results less convincing.
3. Lack of enough quantitative analysis showcasing performance improvements on image-to-3d frameworks.
- Image-to-3D models can use images rendered from original 3D objects as input and ground truth, and then use the metrics of 2D images for measurement, such as SSIM, PSNR, etc. The GSO dataset or Objaverse dataset can be used.
4. The impact of methods on training time is not introduced.
- The paper refine 3D contents generated by baseline models, but the impact of training time is not addressed.

**Questions:**

1. What does consistency refer to in the paper? The SDS-based method is optimized to generate a complete 3D content, not like directly using 2D diffusion models to generate multi-views. Does inconsistency refer to Janus problem?
2. Why need an additional $L_{cg}$ loss? Why not add negative prompts directly to the 2D diffusion models in the top left of Figure 3 for $L_{SDS}$ loss?
3. Is it necessary to first use the baseline methods to obtain a raw 3D content before cooperating with Multi-modal LLM for subsequent operations.

**Limitations:**

The paper has addressed the potential negative societal impact: the potential misuse of advanced 3D generation technologies could undermine social trust and compromise information integrity. And there are no other obvious potential social impacts.

---

> ### Author Rebuttal · Authors · 2024-08-07
>
> Thank you for your feedback. We appreciate your recognition of our method's innovation and effectiveness. Here are our responses to your comments.
>
> **W1. "In lines 148-150, the paper claims defining a new denoising strategy, but there's no further details. Lines 184-186 provide some description, but lack a detailed explanation of how to utilize appearance attention."**
>
> A1. Thank you for pointing this out. We’d like to clarify that the "new denoising strategy" mentioned in lines 148-150 refers to the introduced strategy in Sec.3.2.
>
> In the denoising strategy $\tilde{\epsilon}_\phi(\cdot)$, $\rm{AAttn}(\cdot)$ in Eq.4 functions as cross-attention. The key and value are derived from the focal view, with each of the four views calculating a distinct query. This ensures that each view aligns its features with the focal view, achieving consistent appearance, and we will make it clearer in further versions.
>
> **W2. "The ablation study lacks quantitative analysis."**
>
> A2. We have added quantitative analysis in Q2 of the Global Review, and the quantitative results further verify the effectiveness of our method.
>
> **W3. "Lack of enough quantitative analysis showcasing performance improvements on image-to-3d frameworks, such as SSIM, PSNR, etc. The GSO dataset or Objaverse dataset can be used."**
>
> A3. Thank you for the suggestion. We followed the experimental setup in [1], randomly selecting 30 objects from both the GSO and Objaverse datasets, totaling 60 objects. To ensure variety, we replaced objects with overly simple structures. We rendered their frontal views at 256x256 resolution for input into our method. Performance was assessed using Chamfer Distance (CD) and Volume IoU (Vol. IoU) for geometric quality, and PSNR, SSIM, and LPIPS for visual quality. The results are shown in the table below.
>
> | Metrics            | DreamGaussian | Hallo3D | Zero-1-to-3 | Hallo3D |
> | :----------------- | :-----------: | :-----: | :---------: | :-----: |
> | CD$\downarrow$     |    0.0185     | 0.0171  |   0.0370    | 0.0283  |
> | Vol. IoU$\uparrow$ |    0.5861     | 0.6099  |   0.4824    | 0.5602  |
> | PSNR$\uparrow$     |    16.502     | 16.518  |   13.433    | 14.930  |
> | SSIM$\uparrow$     |    0.8543     | 0.8793  |   0.7210    | 0.7527  |
> | LPIPS$\downarrow$  |    0.2025     | 0.1726  |   0.3926    | 0.3328  |
>
> The experimental results show that our method outperforms the baseline across all metrics, improving both geometry and textures. This further confirms the broad applicability of our approach, enhancing both text-to-3D and image-to-3D tasks.
>
> **W4. "The impact of methods on training time is not introduced."**
>
> A4. We measured the time overhead for both 3DGS-based and NeRF-based baselines, and the results show minimal additional time costs, as detailed in Q1 of the Global Review.
>
> **Q1. "What does consistency refer to in the paper? The SDS-based method is optimized to generate a complete 3D content, not like directly using 2D diffusion models to generate multi-views. Does inconsistency refer to Janus problem?"**
>
> A-Q1. Yes, inconsistency refers to the Janus problem. The term consistency refers to a 3D object maintaining its normal visual structure and appearance across multiple views.
>
> **Q2. "Why need an additional $\mathcal{L} _ {\rm{CG}}$ loss? Why not add negative prompts directly to the 2D diffusion models in the top left of Figure 3 for $\mathcal{L} _ {\rm{SDS}}$ loss?"**
>
> A-Q2. Thanks for your suggestion. Indeed, adding negative prompts directly to the 2D diffusion models is what we do in the w/o C configuration. We also introduced an ablation version, excluding the $P _ E ^ -$ output by LMM from the $\mathcal{L} _ {\rm{CG}}$ calculation, to isolate $P _ E ^ -$ impact. The outcomes from both ablation versions thoroughly evaluate the role of $\mathcal{L} _ {\rm{CG}}$, confirming its necessity.
>
> **Q3. "Is it necessary to first use the baseline methods to obtain a raw 3D content before cooperating with Multi-modal LLM for subsequent operations?"**
>
> A-Q3. Yes. We initially employ baseline methods to generate a raw 3D content. The method we proposed is designed to perform hallucination detection and mitigation on the results produced by the 3D baseline. Therefore, it's necessary to have 3D content prior to the detection and correction process.
>
> ---
> We hope our response is helpful in addressing your concerns. We look forward to continuing our communication with you.
>
>
> [1] Long X, et al. Wonder3d: Single image to 3d using cross-domain diffusion. In *CVPR*, 2024.

---

> > ### Comment · Reviewer_bBV6 · 2024-08-12
> >
> > Thank you to the author for the response. I have two additional questions:
> >
> > 1. Upon reviewing the quantitative results of the ablation study, it appears that the addition of self-attention did not result in significant changes. What might be the reason for this?
> >
> > 2. Could you analyze why incorporating the output of the LMM into the training of the SDS Loss is less effective than handling the CG Loss and SDS Loss separately?

---

> > > ### Author Response · Authors · 2024-08-12
> > > **Official Comment by Authors**
> > >
> > > We sincerely appreciate your thoughtful feedback on our rebuttal and the opportunity to further clarify our points. Thank you for your continued engagement with our work.
> > >
> > >
> > > **Q1: Upon reviewing the quantitative results of the ablation study, it appears that the addition of self-attention did not result in significant changes. What might be the reason for this?**
> > >
> > > CLIP-Score is widely used to estimate 3D consistency [1, 2, 3]. In line with these approaches, our ablation study reports the average CLIP-Score, calculated from the CLIP-Scores between 3D rendered images from different viewpoints and the prompt. However, due to the complexity of 3D generation, no single metric can fully capture consistency [2]. Consequently, the CLIP-Score may not completely account for appearance-related issues, such as color discrepancies, texture variations, or blurry halos across different viewpoints. While the images may still align with the prompt, these subtle appearance details may not be fully reflected by the CLIP-Score. For instance, differences between B/16 and L/14 highlight the role of A, but this difference is less noticeable in B/32.
> > >
> > > To address this, we drew on the metric design from [4] by calculating the LPIPS between adjacent viewpoints and averaging these values across all viewpoints to compute the A-LPIPS. LPIPS offers a perceptual similarity measure that is closer to human vision, and we believe this metric better reflects the impact of the "Multi-view Appearance Alignment" module in our work. The experimental results are as follows:
> > >
> > > | Metrics           | Hallo3D | w/o A  | w/o B  | w/o C  | w/o C & $P ^ {-} _ E$ | Baseline |
> > > | ----------------- | :-----: | :----: | :----: | :----: | :-------------------: | :------: |
> > > | A-LPIPS$\uparrow$ | 0.1863  | 0.1709 | 0.1582 | 0.1479 |        0.1382         |  0.1237  |
> > >
> > > It can be seen that module A plays a crucial role in Hallo3D, as its absence leads to a noticeable performance drop. This is also illustrated in Fig. 7. In the left part, the lion in the second row appears dim in the first column but much brighter in the last four columns, indicating some inconsistencies in lighting. On the right, the deer in the second row shows a blurry halo, and the third column displays more saturated colors than the fourth, suggesting some distortion.
> > >
> > > As discussed, the "Multi-view Appearance Alignment" module effectively enhances 3D consistency. In future work, we are open to collaborating with the research community to develop better metrics for evaluating 3D consistency and improving 3D generation methods.
> > >
> > > **Q2: Could you analyze why incorporating the output of the LMM into the training of the SDS Loss is less effective than handling the CG Loss and SDS Loss separately?**
> > >
> > > $\mathcal{L} _ {\rm{CG}}$ involves multiple denoising steps, which increases its ability to correct inconsistencies, making it more effective in addressing issues identified by $P_E^-$. In contrast, $\mathcal{L} _ {\rm{SDS}}$ focuses on overall generation quality, using single-step denoising to fit the diffusion model’s distribution. Therefore, we incorporate $P _ E ^ -$ into the calculation of $\mathcal{L} _ {\rm{CG}}$ to more effectively enhance 3D consistency.
> > >
> > >
> > >
> > >
> > > [1] Yi T, et al. Gaussiandreamer: Fast generation from text to 3d gaussian splatting with point cloud priors. In *CVPR*, 2023.
> > >
> > > [2] Liu F, et al. Sherpa3d: Boosting high-fidelity text-to-3d generation via coarse 3d prior. In *CVPR*, 2024.
> > >
> > > [3] Tang J, et al. Dreamgaussian: Generative gaussian splatting for efficient 3d content creation. In *ICLR*, 2023.
> > >
> > > [4] Susung Hong, et al. Debiasing Scores and Prompts of 2D Diffusion for View-consistent Text-to-3D Generation. In *NeurIPS*, 2023.

---

> > > > ### Comment · Reviewer_bBV6 · 2024-08-13
> > > >
> > > > Thanks for the response! I don't have further questions and I raise my grade to 5.

---

> > > > > ### Author Response · Authors · 2024-08-13
> > > > > **Official Comment by Authors**
> > > > >
> > > > > Thank you for your recognition and encouragement! We will take your suggestions into account and improve our paper in the further version.

---

> > > ### Author Response · Authors · 2024-08-12
> > > **Official Comment by Authors**
> > >
> > > We sincerely hope our response has addressed your concerns, and we genuinely look forward to further communication with you!

---

### Official Review · Reviewer_gwDu · 2024-07-17

**Soundness:** 3
**Presentation:** 3
**Contribution:** 3
**Rating:** 6
**Confidence:** 5

**Summary:**

This paper aims to alleviate the multi-Janus problem in SDS-based 3D generation tasks. Inspired by the spatial structure inference capability of large multimodality models (LMMs), they propose a novel automatic negative prompt strategy. Specially, they input rendered images and 3D-aware inquiry prompts to LMM to obtain negative prompts. To keep the semantic consistency, they regenerate rendered image guided by a negative prompt and calculate regularization loss between the originally rendered image and the regenerated one.

**Strengths:**

- A novel and interesting strategy to introduce LMM automatically generating negative prompts to alleviate Janus issue.
    - Different from direct use generated negative prompts, this work introduces a prompt-enhanced re-consistency scheme that regenerates render image guided by negative prompt and calculates an MSE loss to help SDS optimization.
- Multi-view Appearance Alignment module calculates self-attention between the focal view of noise x and others to introduce appearance consistency across different views.
- The proposed method is applied to various baselines, including different 3D representations (NeRF and Gaussians) and tasks (text-to-3D and image-to-3D), to show robustness and generalization ability.

**Weaknesses:**

- Lack of detaild of implementation and ablation study. I doubt the reproduction of this work.
    - For Multi-modal Hallucination Detection, is P_I  always the same as the prompts in Figure 4? All rendered images are inputted to LMM?   Which LMM is used in the final experiments, given GPT-4V and LLaVA in Figure 4?
    - For Prompt-Enhanced Re-consistency, how to balance loss_SDS and loss_CG? Can you provide the loss curve of them? The author mentions that this module only works when the rendered images exhibit complete semantic structures. So what is the proportion in the training does it work on average? Why Dψ can produce "None"? I cannot find the corresponding instruction in Figure4's prompts.
    - For the Ablation Study, does w/o B mean that there are no adaptive negative prompts and a general negative prompt is used in the Prompt-Enhanced Re-consistency module to calculate L_CG? So what is the general negative prompt in w/o B ablation study? In w/o C study, does this mean that the generated negative prompt is directly used in equation (3) and no L_CG?

- Lack of some necessary comparison.
    - Please show the training time compared with multiple baselines. In my aspect, inference LMM for each iteration may introduce more training time.
    - The ablation study and analysis are not complete. It's interesting to show that replacing P_E^− with a general negative prompt. And why not introduce L_CG all the time? What is the impact of P_I for Hallucination Detection?
    - Magic3D is very similar with DreamFusion-IF except for the additional finetune phrase. Actually, I think finetune stage has less impact for geometric structures. I suggest authors replace Magic3D with ProlificDreamer. Prolificdreamer has a serious Janus problem but it produces clearer texture which is beneficial for Multi-modal Hallucination Detection.
    - Lack of comparison with mentioned prompt engineering methods: "prep-neg" and "debiasing scores and prompts".

**Questions:**

- How to select a focal view in the Multi-view Appearance Alignment module is unclear. Is the focal view chosen randomly? Or the front view as a focal view always be chosen for each iteration?
- Why the shown examples are low quality? Can the proposed method work for high-quality 3D generation?
- Why only provide one 360-degree example? Why not consider providing complete results in an appendix?

**Limitations:**

Based on the provided examples, it seems like this work just works for very low-quality 3D generation and simple prompts.

---------------------

This work presents the integration of VLM into text-to-3D generation, framing the Janus problem as hallucination detection. It automatically produces adaptive negative prompts via VLM. While the effect is constrained, this approach, devoid of training and 3D priors, offers insights into leveraging LLM in 3D generation. The primary constraint lies in the additional time overhead.

---

> ### Author Rebuttal · Authors · 2024-08-07
>
> Thank you for your feedback. We appreciate your recognition of our method's innovation and applicability. Here are our responses to your comments.
>
> **W1. "I doubt the reproduction of this work."**
>
> Thanks for pointing this out. We'll share our code upon publication to help with reproducing our work.
>
> **W1.1-1. "Is P_I always the same as the prompts in Figure 4?"**
>
> A1.1-1. We used different P_I than in Fig.4. The primary purpose of Fig. 4 is to use a case study to demonstrate how LMMs can infer structural consistency and respond in specific formats. To highlight this capability, we employed two dialogues. In practice, we used a single interaction to query the LMM to achieve faster runtime. Please refer to Q4.1 in the Global Review for the specific setting of P_I.
>
> **W1.1-2. "All rendered images are inputted to LMM?"**
>
> A1.1-2. For efficiency, we input only one of the four rendered images into the LMM, selecting it randomly to avoid over-intervening in a specific view.
>
>  **W1.1-3. "Which LMM is used?"**
>
> A1.1-3. For LMM, we chose the locally deployed LLaVA, using the version llava-v1.6-34b.
>
> **W1.2-1. "How to balance loss_SDS and loss_CG? "**
>
> A1.2-1. Thanks for your pointing this issue. There's actually a typo in Eq.8. To balance L_SDS and L_CG at the same order of magnitude, we set w=0.1  for L_SDS. We'll correct this in the further version. Here's the correct formula:
>
> $$
> \mathcal{L}(\theta) = \mathcal{L} _ {SDS} + w\mathcal{L} _ {CG}, \text{\quad  if } D _ \psi(x, P_I) \text{ is not None}
> $$
> And the other case is:
>
> $$
> \mathcal{L}(\theta) = \mathcal{L} _ {SDS}, \text{\quad  if } D _ \psi(x, P_I) \text{ is None}
> $$
>
> **W1.2-2. "Can you provide the loss curve of them?"**
>
> A1.2-2. Please refer to Q3 of the Global Review for the detail.
>
> **W1.2-3. "What is the proportion does L_CG work?"**
>
> A1.2-3. We provided quantitative ablation results in Q2 of the Global Review. The analysis indicates that adding L_CG increases the average CLIP-Score from 24.32 to 27.03, contributing significantly to the overall improvement. Additionally, the CLIP-Score curve in Fig.4 (PDF) further demonstrates the effectiveness of L_CG.
>
> **W1.2-4. "Why Dψ can produce 'None'?"**
>
> A1.2-4. When Dψ receives low-quality or poorly angled images, LMMs may fail to generate the expected negative prompt. The output then cannot be recognized by our regex and returns None. This process is visualized in Fig.5 in the PDF.
>
> **W1.3. "About general negative prompt in w/o B and w/o C."**
>
> A1.3. w/o B means that P_E^- is not involved in L_CG, while w/o C indicates that P_E^- is involved in L_SDS during the ablation of L_CG. Our method acts as a universal enhancement for 3D generation, considering the common use of general negative prompts in baseline methods[1]. The general negative prompt used in w/o B can be found in Q4.2 of the Global Review.
>
> **W2.1. "About the training time."**
>
> A2.1. We detailed the training time in Q1 of the Global Review. LMM does introduce additional training time, but we believe this is acceptable given the significant improvement in performance.
>
> **W2.2-1. "About the ablation."**
>
> A2.2-1. More ablation study and analysis are detailed in Q2 of the Global Review.
>
> **W2.2-2. "Replacing P_E^− with a general negative prompt?"**
>
> A2.2-2. As discussed in A1.3, we adopt a generative negative prompt (which is used by baseline methods) in all the ablation study.  Specifically, w/o B removes  P_E^− and kept general negative prompt.
>
> **W2.2-3. "Introduce L_CG all the time?"**
>
> A2.2-3. L_CG is designed to constrain the view consistency, but in the early stage, when image quality is too low and lacks sufficient structural information, constraining with L_CG is less meaningful. Additionally, applying L_CG only in the later stages can also reduce training overhead.
>
> **W2.2-4. "What is the impact of P_I for Hallucination Detection?"**
>
> A2.2-4. The two dialogues in Fig.4 correspond to the two roles of P_I:
>
> 1. Activating the LMM's ability to infer 3D view consistency. By including query necessary information about inconsistencies in P_I, the LMM is prompted to identify specific issues in the image.
> 2. Standardizing the LMM's output format. By providing a one-shot example, the LMM's output is guided to match the negative prompt format, making it easier for our designed regular expression to recognize.
>
> **W2.3. "Replace Magic3D with ProlificDreamer."**
>
> A2.3. Thanks for suggestion. We added experiments on ProlificDreamer. The results in Fig.2 (PDF) show our method can improve its consistency.  We will include more detailed discussion about  ProlificDreamer in the final version.
>
> **W2.4. "Comparison with 'perp-neg' and 'debiasing scores and prompts'."**
>
> A2.4. Thanks for the suggestion. We added experiments on Perp-Neg and Debiasing Scores and Prompts. Results in Fig. 3 (PDF) show that our method improves consistency better than both approaches, as reflected in the quantitative and qualitative outcomes.
>
> **Q1. "How to select a focal view?"**
>
> We use Fovy, the camera's vertical view, as our selection standard. The first view with Fovy exceeding 120% of each baseline's default becomes our focal view, enabling a broader shot capturing more object details.
>
> **Q2. "Why the shown examples are low quality?"**
>
> A-Q2. Our method aims to boost view consistency in 3D generation, with quality hinging on the baseline models. We've tested this on high-quality models like GaussianDreamer,  DreamGaussian, and ProlificDreamer.  The results confirm our method's versatility, elevating both lower and higher-quality 3D generations.
>
> **Q3. "Provide 360-degree example?"**
>
> A-Q3. Thanks for suggestion. We have displayed the 360-degree view of the results in Fig.1 (PDF).
>
> ---
> We hope our response resolves your concerns. Due to character limitations, we've provided a concise answer to your questions. We look forward to further communication with you!
>
>
> [1] Yuan-Chen Guo, et al. Threestudio. https://github.com/threestudio-project/threestudio, 2023.

---

> > ### Comment · Reviewer_gwDu · 2024-08-09
> >
> > Thanks for the detailed implementation supplement provided in response, addressing concerns regarding reproducibility. Kindly incorporate these details into the updated Appendix. And supplementary experiments showcasing Hallo3D's superior CLIP Score and the loss curve trends are beneficial for understanding the effectiveness of this work.
> >
> > **Questions Raised in the Rebuttal**
> >
> > **Q1: Time Consumption**
> >
> > Please clarify that "Original Time" corresponds to the baseline method with 1200/2500 iterations. Concern arises over the substantial time consumption introduced by Hallo3D, particularly concerning the potential increase with high-resolution generation. Additionally, it is queried whether all results are produced with training solely 1200/2500 iterations for Gaussian and NeRF representations, a difference from conventional implementation in 3D generation.
> > Insufficient training may impact the quality of results in the paper. Addressing how to achieve high-quality results within a reasonable time is important.
> >
> > **Confusion in Fig.2**
> >
> > Please explain the suspected dual-headed appearance observed in the second render image of Hallo3D in rebuttal Fig.2. Furthermore, for enhanced difference, consider changing the prompt in rebuttal Fig. 3 in the updated version, as prompts like "cottage" typically do not exhibit Janus issues in 3D generation.

---

> > > ### Author Response · Authors · 2024-08-09
> > > **Official Comment by Authors**
> > >
> > > Thank you for taking the time to read our rebuttal and for providing your thoughtful response! We appreciate the opportunity to address these further concerns and provide additional clarification.
> > >
> > > **Q1: Time Consumption**.
> > >
> > > **Q1.1: About the substantial time consumption.**
> > >
> > > A1: Thank you for your suggestion. We will clarify the "Original Time" and the corresponding training iterations in future versions. While our method does introduce additional time overhead, we believe it is within an acceptable range given the improvements in performance and quality it provides, especially considering the challenging nature of addressing the Janus Problem.
> > >
> > > **Q1.2: About the training iterations.**
> > >
> > > Since our method includes the "Multi-View Appearance Alignment" module, which requires attention calculations across **four** differently angled rendered images, we set the batch size to 4 for all baselines. To ensure a fair comparison, we reduced the number of iterations to 1/4 of the original. For example, DreamFusion originally trained for 10,000 iterations, and we adjusted it to 2,500 for optimization. GaussianDreamer(iteration=1200) already uses batch=4, so we matched its iteration count at 1,200.
> > >
> > > Additionally, as shown in Fig. 4, GaussianDreamer has already converged at 1,200 iterations, indicating sufficient training. Empirically, we also observed that further increasing the number of iterations did not improve the consistency of 3D generation.
> > >
> > > **Q2: Confusion in Fig.2**.
> > >
> > > A2: Thank you for your careful observation. Completely resolving the Janus Problem remains a significant challenge in the field. Our method specifically aims to enhance view consistency in 3D generation by addressing the hallucination issues commonly found in large models. As demonstrated by both quantitative and qualitative experiments, while there may still be a small presence of inconsistencies, our approach effectively mitigates these issues.
> > >
> > > As noted in W2.3, ProlificDreamer produces impressive 3D quality but also exhibits a pronounced Janus Problem. In our experiments using ProlificDreamer as a baseline, we ensured a genuine evaluation of our method's performance by not cherry-picking any results. Despite this, as seen in Fig.2, our method still shows significant improvements over the baseline, greatly enhancing consistency.
> > >
> > > Our method has demonstrated improvements in 3D generation consistency across various baselines (both text-based and image-based). Moving forward, we will treat the Janus Problem as a key research direction and look forward to contributing further to 3D generation alongside our fellow researchers.
> > >
> > > Due to the limitations of the rebuttal format, we regret that we are unable to further modify the PDF to provide a more typical prompt visualization. However, we will modify the "cottage" prompt to better present a typical Janus prompt setup in final version.
> > >
> > > ---
> > >
> > > Thank you once again for your valuable suggestions. We hope our response has addressed your concerns and would be happy to continue the discussion with you!

---

### Author Rebuttal · Authors · 2024-08-07

We thank the reviewers for their constructive feedback and valuable insights, which have significantly contributed to the improvement of our research. We are grateful for your thoughtful suggestions.

Our work has been recognized for ***the innovative introduction of the LMMs*** (gwDu, bBV6, isRS), ***the broad applicability*** (gwDu, bBV6), ***the significant performance improvements*** (bBV6, isRS), ***the solid comparative experiments*** (isRS), and ***the clear method description*** (isRS).

**Q1. "The time consumption introduced by Hallo3D."**

**A common concern is the additional time consumption introduced by Hallo3D**, particularly regarding the extra inference time for LMMs. To address this concern, we recorded the runtime using [1] (*based on 3DGS with fewer iterations and faster speed*) and [2] (*based on NeRF with more iterations and slower speed*) as baselines, on NVIDIA V100.

**Specifically, we begin calculating $\mathcal{L} _ {\rm{CG}}$ later in the training process and do so every 4 iterations in our experiments.** This approach aligns with the statement in Sec 3.4 that *"this module only works when the rendered images exhibit complete semantic structures."* The rationale is twofold. First, in the early stages of training, the 3D assets are relatively disorganized and lack clear semantic structures, making it difficult for LMMs to reason accurately. Therefore, we introduce $\mathcal{L} _ {\rm{CG}}$ later in the training. Second, we empirically found that calculating $\mathcal{L} _ {\rm{CG}}$ every 4 iterations does not impact performance, so we adopted this approach to reduce training time. The results are shown in the table below.

| Baseline            | Iteration | $\mathcal{L} _ {\rm{CG}}$ Start Rounds | Original Time | Extra Time | Total Time |
| :------------------ | :-------: | :------------------------------------: | :-----------: | :--------: | :--------: |
| GaussianDreamer [1] |   1200    |                  1000                  |    ~28 min    |  ~10 min   |  ~38 min   |
| DreamFusion[2]      |   2500    |                  2200                  |    ~51 min    |  ~15 min   |  ~66 min   |

Combining the table above with Fig.4 in the PDF, it can be seen that **Hallo3D efficiently enhances cross-view consistency with acceptable time costs**.

**Q2. "The quantitative ablation experiment and analysis."**

**We conducted quantitative ablation experiment** to further demonstrate the necessity of each module. Identical to the setup in paper, module A represents Multi-view Appearance Alignment, module B stands for Multi-modal Hallucination Detection, and module C denotes Prompt-Enhanced Re-Consistency. Additionally, we included experiments for the scenario "w/o C & $P_E^-$", where without providing $P ^ {-} _ E$ for $\mathcal{L} _ {\rm{SDS}}$ when w/o C. The results are as follows.

| Metrics                   | Hallo3D | w/o A | w/o B | w/o C | w/o C & $P ^ {-} _ E$ | Baseline |
| ------------------------- | :-----: | :---: | :---: | :---: | :-------------------: | :------: |
| CLIP-Score B/32$\uparrow$ |  24.25  | 23.98 | 23.65 | 22.46 |         22.23         |  21.27   |
| CLIP-Score B/16$\uparrow$ |  26.83  | 25.88 | 25.10 | 23.59 |         23.23         |  22.67   |
| CLIP-Score L/14$\uparrow$ |  30.00  | 29.36 | 28.71 | 26.92 |         25.58         |  23.71   |

The better performance of "w/o C" compared to "w/o C & $P ^ {-} _ E$" also supports the necessity of introducing $\mathcal{L} _ {\rm{CG}}$.


**Q3. "The loss curve between $\mathcal{L} _ {\rm{SDS}}$ and $\mathcal{L} _ {\rm{CG}}$."**

The curves are shown in Fig.4 of the PDF. Additionally, we've also included the CLIP-Score for the full model and the ablated version without $\mathcal{L} _ {\rm{CG}}$. It can be observed that $\mathcal{L} _ {\rm{CG}}$ decreases with the number of iterations and significantly improves the CLIP-Score, whereas the CLIP-Score without $\mathcal{L} _ {\rm{CG}}$ shows only a minor improvement, demonstrating the effectiveness of $\mathcal{L} _ {\rm{CG}}$.

**Q4. "The Prompt settings."**

**Q4.1. "The $P_I$ setting in LMM."**

> *"You are a master of 3D generation, and please refer to the 'Prompt' and 'Negative Prompt' below to identify the inconsistency in the image I provided you with, with body shape, perspective, texture, and so on.*
>
> *Reference:*
>
> *'Prompt': '3d render of xx, front view, standing, high quality, 4K',*
>
> *'Negative Prompt': 'multi-head, unnatural lighting, smooth appearance, distorted color, long neck,  two-nosed, extra limbs' ".*

**Q4.2. "The general negative prompt setting in LMM."**

> "unnatural colors, poor lighting, low quality, artifacts, smooth texture".

---

We believe that Hallo3D can be a valuable supplement to the NeurIPS community, particularly with the enhancements made based on the reviewers' feedback, which have helped us better convey the effectiveness of our method.

Thank you! The Authors.


[1] Taoran Yi, et al. GaussianDreamer: Fast Generation from Text to 3D Gaussians by Bridging 2D and 3D Diffusion Models. In *CVPR*, 2024.

[2] Ben Poole, et al. DreamFusion: Text-to-3D using 2D Diffusion. In *ICLR*, 2022.

---

### Decision · Program_Chairs · 2024-09-25

**Decision:**

Accept (poster)

**Comment:**

In the initial reviews, all reviewers acknowledged the innovative introduction of the LMMs, the significant performance improvements, and the solid comparative experiments. However, some reviewers pointed out the missing quantitative ablation experiment and analysis, and confused term such as consistency in the paper. During the rebuttal phase, the authors addressed these concerns with additional experiments, leading to positive feedback from all reviewers. AC has also carefully reviewed the paper, the reviews, and the rebuttal, and concurs with the reviewers' opinions. Therefore, AC recommends acceptance. It would be highly beneficial to include the additional results from the rebuttal phase in the final version.